# Beyond bNAbs: Uses, Risks, and Opportunities for Therapeutic Application of Non-Neutralising Antibodies in Viral Infection

**DOI:** 10.3390/antib13020028

**Published:** 2024-04-03

**Authors:** Kahlio Mader, Lynn B. Dustin

**Affiliations:** Kennedy Institute of Rheumatology, University of Oxford, Roosevelt Drive, Headington, Oxford OX3 7FY, UK; kahlio.mader@st-annes.ox.ac.uk

**Keywords:** immunoglobulin, viral infection, neutralizing antibodies, antibody engineering, antiviral immunity, Fc receptors

## Abstract

The vast majority of antibodies generated against a virus will be non-neutralising. However, this does not denote an absence of protective capacity. Yet, within the field, there is typically a large focus on antibodies capable of directly blocking infection (neutralising antibodies, NAbs) of either specific viral strains or multiple viral strains (broadly-neutralising antibodies, bNAbs). More recently, a focus on non-neutralising antibodies (nNAbs), or neutralisation-independent effects of NAbs, has emerged. These can have additive effects on protection or, in some cases, be a major correlate of protection. As their name suggests, nNAbs do not directly neutralise infection but instead, through their Fc domains, may mediate interaction with other immune effectors to induce clearance of viral particles or virally infected cells. nNAbs may also interrupt viral replication within infected cells. Developing technologies of antibody modification and functionalisation may lead to innovative biologics that harness the activities of nNAbs for antiviral prophylaxis and therapeutics. In this review, we discuss specific examples of nNAb actions in viral infections where they have known importance. We also discuss the potential detrimental effects of such responses. Finally, we explore new technologies for nNAb functionalisation to increase efficacy or introduce favourable characteristics for their therapeutic applications.

## 1. Introduction

Virus-neutralising antibodies (NAbs) refer to those that directly impair viral entry into target cells, blocking the initiation of the viral life cycle [1]. Neutralising activity is typically mediated by the Fab (Fragment antigen-binding) portion of immunoglobulin binding to the viral surface in such a way that it blocks engagement of host entry receptors [2,3]. Such neutralisation may also include Fc-mediated steric hindrance [4,5]. In addition to extracellular neutralisation, more recent reports explore the concept of intracellular neutralisation following antibody or antibody-antigen complex internalisation, including in endosomes [3,6]. In contrast, non-neutralising antibodies (nNAbs) can also mediate humoral protection [6].

nNAbs can mediate protection through Fc (Fragment crystallisable)-mediated functions, such as innate immune cell activation, chemokine and cytokine release to regulate immune cell activation and recruitment [7], antibody-dependent cytotoxicity (ADCC), antibody-dependent cellular phagocytosis (ADCP) and antibody-mediated complement deposition (AMCD) (Figure 1). ADCC refers to the activation of innate immune cells, notably NK cells, to release cytotoxic granules containing perforin and granzyme. This process depends on NK cell FcγRIIIA engagement by Fc following antibody recognition of antigen on the surface of infected cells [8,9]. These cytotoxic granules form pores in the target cell membrane and activate caspases, leading to mitochondrial dysfunction and subsequent apoptosis of the infected cell. ADCP similarly relies on the interaction of Fc with Fc receptors (FcRs) expressed on phagocytic cells such as macrophages. This engagement results in the internalisation of the antibody-opsonised viral particles and their degradation within the formed phagolysosome [10]. Depending on the specific cell type that engulfs the opsonised viral particle, this also initiates further antiviral defences via increased immune activation through antigen presentation by macrophages or dendritic cells or cytokine release by plasmacytoid dendritic cells [10]. Neutrophils are also potent phagocytes and inducers of cytotoxicity through reactive oxygen species production [11]. In addition, neutrophils can undergo a specific form of cell death termed NETosis, where extracellular traps are expelled from the neutrophil to capture and degrade extracellular viral particles [11]. In AMCD, through the classical pathway, opsonising antibodies bind C1q molecules, initiating an amplification cascade of C3 convertase activation and C3 production before C5 generation by C5 convertase [12]. These, along with other complement components, assemble to form the membrane attack complex, which, analogous to ADCC, results in membrane permeabilization and induction of apoptosis of the opsonised virally infected cell [12].

As the Fc domain mediates antibody effector functions, these functions are isotype dependent. For example, Fcγ receptors possess a low affinity for IgM, which, therefore, does not readily induce ADCC [13], an important non-neutralising effector mechanism of IgG [14]. Despite the typically lower affinity of IgM for antigen relative to IgA or IgG (due to lack of class switching or affinity maturation [15]), IgM antibodies possess high avidity and exhibit potent complement-dependent cytotoxicity (CDC) through C1q binding, both features resulting from their pentameric structure [13]. IgA does not express this C1q binding site, so unlike IgM or IgG, classical complement activation by IgA is not expected [16]. Furthermore, recent studies have highlighted the ability of dimeric IgA to bind cytosolic proteins following cell entry via transcytosis [17]. IgA may thus function intracellularly [3] as part of its known roles in mucosal antiviral defence [18,19,20].

As such, nNAbs can be vitally important to host antiviral responses. nNAb functions can be identified through both targeted and non-targeted approaches but, by definition, are missed by neutralisation assays. Whilst the vast majority of antibodies generated against a virus are non-neutralising [21], this does not rule out protective functions. Antibodies do not necessarily target entry receptors or fusion proteins that are commonly involved in direct neutralisation [22]. They may often target other structural or even non-structural proteins, with non-structural proteins involved in virus replication and assembly but not incorporated into the virion itself [23]. Other structural proteins, such as nucleocapsids, do not have a role in virus attachment to target cells but are instead incorporated into the viral particle for other purposes [24].

Whilst NAbs typically bind highly variable epitopes as the virus undergoes mutation of these sites to evade host immunity, broadly neutralising antibodies (bNAbs) typically target conserved indispensable regions, enabling their high breadth of activity [25,26,27]. This property has sparked great interest in bNAbs therapeutically and underlies why bNAbs are the gold standard for vaccine-induced humoral responses [28,29,30,31,32]. Similarly, nNAbs may target conserved regions; however, binding to these sites does not directly inhibit viral infectivity but can still facilitate protection [33,34,35]. In addition to direct prevention of viral entry into cells, antibodies mediate myriad effector functions through Fc domain interactions. To this end, nNAbs are important immune effectors even in the absence of direct neutralisation. This review will discuss the importance of nNAbs, both their protective and harmful effects as well as the potential for their functionalisation.

## 2. Antibody Activities Beyond Neutralisation

Extracellular neutralisation is typically the metric of activity of NAbs, for example, through the uptake of labelled viruses by target cells in in vitro neutralisation assays. Additional activities have been identified beyond extracellular neutralisation and even Fc-effector functions. These activities include blocking of virus assembly and release and inhibition of viral replication through intracellular activities. Using fluorescence imaging, He et al. demonstrated that antibodies targeting hemagglutinin (HA), neuraminidase, and M2 could all inhibit egress and release of influenza A viruses from influenza-infected HEK-293T and MDCK-II cell lines [36]. This was assessed through the treatment of cells with antibodies following virus entry. The activity was associated with crosslinking of viral protein targets, either in cis (within one cell or viral particle) or in trans (between the cell and the viral particle). Even antibodies against M2, which is not highly exposed on the viral particle and does not mediate cell entry, were effective [36]. A crosslinking mechanism is also supported by old literature, where bivalent anti-M2 antibody 14C2 inhibited influenza particle release, but the monovalent Fab did not [37].

Phanthong et al. used antibodies specific to the enterovirus capsid protein, VP4, to inhibit viral replication. These reagents included scFvs (single-chain variable fragments) and transbodies (cell-penetrating antibodies) lacking Fc domains [38]. EV71-infected cells were treated with scFvs and transbodies specific for VP4, which resulted in reduced vRNA release, reduced infectivity of released virions, and increased host cell innate antiviral responses [38]. Of note, VP4′s inner capsid location makes it unlikely to be accessible to NAbs [39,40]. The targeted VP4 domain plays important roles in membrane pore formation and viral genome release, and anti-VP4 reagents may inhibit these VP4 functions [38]. As many viruses, such as HIV and influenza virus, also increase cell membrane permeability, scFv and transbody-based intracellular therapy strategies could possibly be developed for these infections [38]. The landscape of potential cell-penetrating antibody therapies in HIV has already been reviewed, with multiple potential viral targets across different stages of the viral lifecycle [41]. These include integration, targeting integrase by sFv-IN, or translation and assembly with scFvs or single domain antibodies against viral components Tat, Nef, Rev and Vif, as well as p24, all developed [41]. Such cell-penetrating antibodies appear more effective than small molecule inhibitors. However, many challenges against their implementation still exist [41]. These include deteriorating bioavailability, high production cost, cytotoxicity and potential emergence of viral resistance [41].

Zhou et al. evaluated the capacity of a novel IgA antibody, 1D11, against a measles virus phosphoprotein to inhibit viral replication [42]. Measles-infected Vero cells were grown in a transwell system and engineered to express poly Ig receptor (pIgR) to enable the uptake of secretory IgA. Whilst 1D11 targets a non-structural protein and thus lacks extracellular neutralising capacity, 1D11-treated cells produced lower viral titres in a manner dependent on pIgR expression [42]. 1D11 may act by interfering with the interaction of the phosphoprotein with viral nucleocapsid during measles virus RNA replication [42].

nNAbs may also activate potent intracellular effector mechanisms. Antibodies to the nucleoprotein of lymphocytic choriomeningitis virus (LCMV) mediate protection via TRIM21, a cytosolic Fc receptor and E3 ubiquitin ligase [43,44]. TRIM21 bound to nNAb-containing immune complexes, targeting N protein for proteasomal degradation and enhancing CD8^+^ T cell responses to the nucleoprotein [43]. This enhancement may be mediated by proteasomal degradation of nucleocapsid and subsequent presentation of nucleocapsid-derived antigenic peptides on class I MHC [43]. TRIM21-dependent recognition of viral antigen-antibody complexes also stimulates innate immune signal transduction, IFN-β transcription, and production of proinflammatory mediators, including chemokines, TNF, and IL-6; it also induces an antiviral state in the absence of extracellular neutralisation [45].

## 3. Protective Functions of nNAbs

The protective capacity of nNAbs has been highlighted across a diverse array of viral infections. This includes an extensive list of animal challenge models in which passively transferred or immunisation-elicited nNAbs reduced viral burden [6]. In Influenza infection, broad protection against infection both prophylactically and therapeutically in vitro and in vivo has been observed [46,47,48,49,50]. Ko et al. (2021) showed that mAb 651, isolated from mice following HA immunisation, was capable of binding the hemagglutinin head region of multiple influenza strains [50]. Whilst in vitro analysis demonstrated no neutralising activity, intraperitoneal injection of mAb 651 following the Influenza challenge significantly improved survival. This effect was lost upon either Fc mutation (preventing Fc receptor binding) or in FcγR knockout mice. This loss of protection was demonstrated to reflect a lack of engagement of NK cell and alveolar macrophage responses [50]. The importance of NK cells in protection was similarly highlighted in a report demonstrating that infusion of IgG Fc regions alone protected mice from lethal HSV1 challenge, reflecting an NK cell-dependent effect independent of Fab targets, with the Fc domain itself mediating recognition of an influenza target [51]. As such, Sedova et al. (2019) [34] argue for the importance of nNAbs in influenza and argue that simply assaying humoral responses through hemagglutinin binding and subsequent neutralisation is insufficient and obsolete for modern vaccines [34]. 

Fujimoto et al. generated transgenic mice expressing human anti-IAV nucleoprotein antibodies derived from an H5N1 avian influenza-infected patient to demonstrate a protective role of IAV nNAbs beyond HA antigens [52]. nNAb expression protected mice from experimental infections with lethal doses of either homologous or heterologous IAV strains. Immunofluorescence demonstrated the capacity of these mAbs to bind nucleoprotein on the surface of infected cells [52]. Whilst not experimentally validated, the nNAbs could conceivably induce Fc effector functions to clear infected cells. Carragher et al. similarly demonstrate the capacity of anti-influenza nucleoprotein antibodies to mediate heterosubtypic immunity in a murine model [53]. This protection was conferred by both vaccination with conserved soluble nucleoprotein or by passive transfer of vaccine serum. Immunisation resulted in modest CD8 T cell responses, suggesting nNAbs may contribute to cytotoxic T cell priming, with authors hypothesising this to reflect cross-presentation [53]. Nucleoprotein-specific responses have also been shown to be important in other viral infections, including murine hepatitis virus, where protection could be conferred through pre-challenge intraperitoneal injection of anti-nucleoprotein mAb [54]. Nucleoprotein is unlikely to be accessible to NAbs in intact murine hepatitis virus, a coronavirus with a lipid envelope surrounding the viral ribonucleoprotein [55]. 

Overlooking the importance of nNAbs and focussing exclusively on neutralisation may also have implications for vaccine regimes against other viral infections. Human papillomavirus (HPV) vaccines have been highly efficacious in decreasing the incidence of HPV infection and cervical cancer worldwide [56,57]. These are typically administered as 2- or 3-dose regimes, yet recent observational studies suggest single-dose regimes to be as protective despite inducing significantly lower NAb titres (reviewed by Quang et al.) [58]. The authors believe this protection may result from the activity of nNAb effector functions such as ADCP [58]. This merits further investigation as, if the immunological basis for these observations can be identified, this could strengthen arguments for the implementation of single-dose regimes vital for low-resource areas.

Fc effector functions have also been shown to contribute to protection in murine SARS-CoV-2 therapeutic models. A non-neutralising mAb to SARS-CoV-2 spike was actually more protective than equivalent doses of a neutralising mAb in human ACE2-transgenic mice when administered one-day post intranasal SARS-CoV-2 infection [59]. The nNAb’s efficacy reflected its opsonic and ADCP activities [59]. In this study, the authors defined equivalent mAb doses based on the differences in affinity between the nNAb and NAb administered [59]. Whilst the study demonstrated the protective capacity of a nNAb, it would have been informative to evaluate whether a modified version of the nNAb that cannot bind to FcRs was similarly protective. However, it has been demonstrated that a nNAb with Fc function-enhancing mutations potentiated the prophylactic efficacy of a NAb in another murine SARS-CoV-2 model [60].

Cytomegalovirus (CMV) is a highly prevalent double-stranded DNA virus that establishes latency, resulting in lifelong infection [61]. Whilst primary infection in otherwise healthy individuals is typically asymptomatic, reactivation in immunocompromised patients can result in organ failure such that the development of vaccines and therapeutics is vital [61]. Whilst NAbs were often superior, prophylactic administration of either neutralising or nNAbs conferred significant protection against lethal MCMV challenge in immunodeficient mice [62]. Human studies have also built upon observations from animal models treated with nNAbs. For example, ADCP and ADCC mediated by maternal non-neutralising IgG Abs corelated with decreased risk of congenital human CMV infection [63,64]. Correspondingly, protection against human CMV infection induced by glycoprotein B vaccination appears independent of NAb induction, yet also seemingly not dependent on ADCC [65]. This is supported by other studies demonstrating that glycoprotein B nNAbs derived from natural human CMV infection facilitate phagocytosis [66].

The roles of nNAbs have been studied in HIV, another latency-establishing infection where nNAbs may mediate protection by several mechanisms [67]. Hioe et al. (2022) administered two non-neutralising human mAbs targeting HIV envelope to humanised mice before challenge with HIV-1; whilst mucosal infection was not blocked, the viral burden was decreased, as measured through plasma viraemia and cell-associated vRNA in tissue [68]. However, measurement of vDNA demonstrated no decrease, suggesting inhibition of viral replication rather than prevention of cellular infection [68]. Protection correlated with FcγRIIA activation, ADCP, and complement fixation. Similarly, the adoptive transfer of purified vaccine-induced polyclonal IgG before infection conferred partial protection in naive SIV-challenged macaques despite the absence of NAb activity [69]. Other reports, such as by Horwitz and colleagues, demonstrate that adoptively transferred nNAbs cleared HIV-infected cells in humanised mice in an Fc-dependent manner; moreover, nNAbs mediated selective pressure in vivo, increasing the frequency of nNAb-resistant viral variants to evade such responses [70]. Correspondingly, in the RV144 HIV vaccine trial, nNAb Fc effector functions were the only identified correlates of protection [71]. Similarly, ADCC breadth and potency of antibodies transferred via breast milk strongly influence mother-to-child HIV transmission [72]. Moreover, in HIV infection in particular, the capacity of IgA and IgM to dimerise or exist as pentamers respectively, means these isotypes can cause viral aggregation, inhibiting viral dissemination with particular attention drawn to IgA1 [73]. 

nNAbs can also synergise with NAbs to mediate a protective effect. For example, the administration of a nNAb isolated from a Marburg virus (MARV) survivor protected mice from the otherwise lethal challenge of mouse-adapted MARV [74]. This was critically dependent on Fc function, with mutations that abrogated FcγR interaction also abrogating the protective effect. Another nNAb assayed potentiated NAb efficacy through increasing accessibility of a neutralising epitope within the receptor binding domain (RBD), as determined in vitro using Jurkat cells expressing surface MARV glycoprotein [74]. Whilst in vivo confirmation of this observation is required in this study, other reports support the role of nNAbs cooperating with sub-neutralising doses of NAb to mediate potent neutralisation against a range of filoviruses, including Ebola and Sudan viruses both in vitro and in mice [75]. This activity was mediated in part by exposure of masked epitopes adjacent to NAb binding sites, where pan-Ebolavirus nNAbs could acquire neutralising capacity when co-administered with sub-neutralising concentrations of NAb to an adjacent epitope [75]. This specific phenomenon has been termed enabling cooperative neutralisation [75]. In some cases, the activity of NAb is critically dependent on Fc effector function. Gunn et al. examined Ebola-specific mAbs with and without neutralising activity to identify Fc features associated with protection [76]. The authors identified a proportion of strong neutralisers to not be protective, while some nNAbs were strongly protective against Ebola virus challenge. Overall, the weaker the NAb activity, the more the protection depended on Fc effector functions; even potent neutralisers were not protective if they were unable to induce phagocytosis [76]. 

Alternatively, Richter et al. demonstrate the capacity of nNAbs against LCMV to protect FcγR or C3 deficient mice against infection [77]. These mice were treated with anti-IL-10 receptor antibodies, demonstrating the capacity for protection even in the absence of IL-10 signalling, Fc effector functions or complement activation [77].

## 4. Detrimental Effects of nNAbs or NAb

On the other hand, there are examples of nNAbs impairing the host antiviral response. Such inhibition can result from nNAbs targeting immunodominant epitopes and masking nearby NAb epitopes, an immune escape mechanism exploited by viruses including hepatitis C and HIV [78,79] (Figure 2). Verrier et al. characterised nine pairs of neutralising and non-neutralising human antibodies to evaluate hindrance against HIV-1 89.6 isolate neutralisation [80]. At the time of this study, the majority of synergy studies were evaluated against T cell line-adapted HIV strains, making these studies inherently less physiologically relevant [80]. The authors developed a mathematical reagent interaction calculation to enable the assessment of anti-gp41 nNAb and NAb combination synergy. Whilst additive effects of certain mAb combinations on neutralisation were observed, with combinations of nNAb 98.6 and NAb 2F5 or 50–69, reduced potency of neutralisation was exhibited. This suggested a form of antagonism by the nNAb. The authors proposed that this effect reflected a mechanism of direct steric hindrance, blocking the NAb binding site, or negative cooperativity where a conformational arrangement induced by nNAb binding occluded a neutralisation epitope [80] (Figure 2). Such competition between NAb and nNAb binding may also occur in SARS-CoV-2 infection and was potentially detected in plasma from COVID-19 patients with severe disease [81].

Alternatively, nNAbs can be deleterious through enhancement of host cell entry, for example, through the engagement of FcR independently of the natural receptor for viral entry [82] (termed ‘extrinsic’ antibody-dependent enhancement (ADE) of infection). Extrinsic ADE is observed when antibodies elicited by primary infection increase the risk of severe disease upon secondary infection with the same or a closely related virus [83,84] (Figure 2). In contrast, intrinsic ADE facilitates enhanced viral replication through suppression of host antiviral responses [85,86] (Figure 2). ADE is of great importance in Dengue virus (DENV) infection, with secondary infection the greatest risk factor for dengue haemorrhagic fever/dengue shock syndrome [87]. However, this is a complex interaction. Katzelnick et al. analysed data from a long-term paediatric cohort repeatedly exposed to DENV [88]. Inhibition ELISA against a mixture of DENV1-4 antigens measured DENV-specific antibody titres and coupled with statistical analysis, demonstrated that low antibody titres did not enhance disease whilst high titres could successfully protect against severe disease. However, an intermediate antibody titre was associated with ADE and severe disease [88]. Thus, this study validated ADE in human DENV infection but also highlighted the nuanced relationship between immune correlates of disease enhancement and those of protection. In particular, prM and fusion loop-specific Abs elicited by Zika virus infection or immunisation have been shown to mediate ADE of Dengue virus infection [89,90]. Likewise, antibodies elicited by Dengue virus immunisation can promote ADE of Zika virus infection in vivo and in vitro, demonstrating the reciprocity [91]. 

Similarly, in vitro studies indicate that complement engagement by patient-isolated nNAbs may facilitate infectivity and dissemination of virus early in HIV infection [92]. This observation was robust across heterologous viral isolates, suggesting the binding of conserved envelope epitopes across strains, although this was not validated [92].

Fc- and FcγRIIA-dependent viral entry of SARS-CoV-2 spike-decorated pseudoparticles has also been demonstrated in vitro in the presence of patient plasma, highlighting a potential role of ADE in SARS-CoV-2 disease [93]. This effect was demonstrated to be titre-dependent, with increasing plasma dilution causing waning neutralisation and an inverse increase in Fc-mediated viral entry. A further interesting observation included significantly lower spike-specific IgM in patient plasma with high Fc-mediated viral entry, suggesting Fc-dependent entry was mediated by another isotype [93]. However, whilst Fc-mediated entry may have been observed, this does not prove ADE. Some reports support the role of ADE in SARS-CoV-2 via a slightly different mechanism than other viruses. Whilst FcγRII**A**-mediated endocytosis by macrophages is well documented for DENV, Wang et al. used pseudovirus infection of Raji and Daudi B cell lines to show FcγRII**B**-mediated ADE of SARS-CoV-2 by human mAbs MW01 and MW05 in vitro [94]. Studies like these, and the observation of increased titres of anti-SARS-CoV-2 antibodies coinciding with greater disease severity, have fuelled debate about ADE in the field [95,96]. However, many studies and reviews do not support a role of ADE in the worsening of SARS-CoV-2 infection [97,98,99].

Shifting of the humoral response to non-neutralising immunodominant epitopes can likewise be deleterious. This is argued to play a role in waning immunity in SARS-CoV-2 infection with adaptation of the memory B cell pool away from spike epitopes to internal viral components such as N protein and ORF8 [100]. This effect is exacerbated in elderly patients, with the spike protein-specific memory B cell population decreased relative to that of younger patients [100]. It is important to identify which characteristics of nNAbs are associated with protection or deleterious effects of evolving humoral responses. In this vein, Chakraborty (2022) identified non-neutralising, afucosylated IgG in patients with severe COVID-19 but not in mRNA-vaccinated patients or those with mild disease [101]. The authors identified afucosylated IgG to be highly inflammatory, as opposed to fucosylated IgG, exhibiting reduced proinflammatory capacity [101]. Further studies dissecting distinct features like these will be vital in understanding whether a nNAb is a friend or foe. 

## 5. Measuring Fc-Dependent Effector Functions

Antigen binding and neutralisation assays do not measure Fc effector functions and are insufficient to determine whether an antibody response is protective. Challenges to assaying Fc effector functions include low assay throughputs, diverse FcRs (with variable signalling capacities and expression levels), and infection-specific impacts on effector cells [102]. Thus, it is imperative to develop robust Fc-effector function screens for their assessment to become commonplace when characterising immune responses. There are a range of commercial kits available to evaluate Fc effector activities, including both in vitro reporter-based assays and those that are primary cell-based. However, issues of consistency in Fc effector evaluation and readouts between experiments and laboratories remain a challenge. 

ADCC can be measured through incubation of antibodies or sera with antigen-bearing target cells and NK cells, followed by evaluation of NK cell degranulation (CD107a) or target cell lysis (flow cytometric viability assays; chromium release assays) [103]. To assay ADCP, antigen-coated fluorescent particles are mixed with antibodies of interest and fed to effector phagocytes; uptake is measured by flow cytometry [102]. Fischinger et al. describe a high-throughput antigen-coated bead-based complement fixation assay to evaluate the potential of antibodies to induce AMCD [104]. Ackerman et al. describe a high throughput ADCP in vitro assay in which antigen-coated fluorescent latex beads are used to capture antibodies of interest, then incubated with a monocytic cell line expressing a wide range of FcRs before flow cytometric measurement of captured beads [105]. Due to the user-selected antigen for bead coating, this assay has the benefit of not requiring prior purification of antibody and the flow format enables simultaneous evaluation of cytokine release via Luminex assay [105]. Similarly, de Neergaard et al. describe a high-sensitivity platform to normalise phagocytosis assays across laboratories and platforms [106]. This method, termed persistent association-based normalisation, employs a dose-response curve-based analysis to standardise cross-laboratory results through standardising measurement and analysis [106]. In a subsequent publication from the group, the authors demonstrate how this can be incorporated into most phagocytosis assays, with consistency of assessment as an important goal for the field [107].

Implementation of Fc effector function assays is exemplified in a study by Natarajan et al. [108], who evaluated convalescent plasma for its antiviral effects on SARS-CoV-2 beyond neutralisation. The authors performed an Fc array assay on antigen-conjugated, convalescent serum-treated microspheres to evaluate Fc features. To evaluate ADCP, they incubated antigen-coated fluorescent microspheres with antibodies and then measured the uptake of the opsonised microspheres by the monocytic cell line THP-1 [108]. Phagocytosis was defined as the uptake of one or more fluorescent beads measured by flow cytometry [108]. A surrogate ADCC assay was performed by measuring FcγRIIIA activation in a Jurkat cell line expressing a luciferase reporter of NFAT activation. After incubation of the antibody sample with RBD-coated wells, the luciferase reporter was measured as a proxy of ADCC potential [108]. AMCD was measured by assaying C3b deposition on antibody-opsonised microspheres [108].

## 6. Exploiting nNAbs for Therapeutic Applications

With continued innovation, suboptimal antibodies can be functionalised or modified to increase their utility as therapeutic agents. These modifications can range from subtype switching to the production of chimeric bi- or tri-specifics to the addition of other biologics or receptors to their structures (Figure 3). Izadi et al. demonstrated the importance of antibody subclass on Fc function and the capacity of nNAbs to be protective [109]. As IgG3 is the most potent activator of the classical complement pathway, they engineered eight non-neutralising IgG1 mAbs against SARS-CoV-2 to bear IgG3 Fc domains. IgG3 isotype conversion enhanced avidity for one mAb, decreased avidity for another, and had no effect on avidity for the other six mAbs. Notably, isotype conversion to IgG3 resulted in a ~2-fold increase in induced phagocytosis of opsonised spike-coated beads by THP-1 cells (used as a proxy for phagocytes). Similarly, when a cocktail of all eight re-engineered nNAbs was evaluated, phagocytic efficiency was increased ~12-fold relative to their IgG1 counterparts, demonstrating that subclass engineering can enhance nNAb protective function [109].

In a similar vein, Weidenbacher et al. also demonstrated the potential utility of functionalised nNAbs for SARS-CoV-2 therapy [35]. Rapid viral evolution has allowed SARS-CoV-2 to outwit many licensed therapeutic anti-spike NAbs, while the capacity of nNAbs targeting conserved regions of SARS-CoV-2 spike has been identified to be preserved across strains [35,110]. Weidenbacher et al. fused a scFv from a nNAb to the N terminus of the ACE2 ectodomain, producing a functionalised antibody whose scFv binds SARS-CoV-2 spike and positions the otherwise low-affinity ACE2 receptor to favour engagement of the RBD. This conformation, therefore, favours non-productive soluble ACE2 binding in preference of host cell surface ACE2, preventing target cell engagement and, therefore, inhibiting viral entry. These ‘Reconnabs’ and bispecific ‘CrossMAbs’ possess the capacity to fully neutralise SARS-CoV-2 in vitro [35]. Thus, we may harness nNAbs with desirable properties, such as the binding of conserved regions, and modify these into novel therapeutic agents.

This strategy of exploiting host receptors has also been applied to other viral infections such as HIV. Richard et al. developed antibody-CD4 chimeric proteins where coreceptor binding site or cluster A-specific IgG1 nNAbs were linked to extracellular domains 1 or 2 of CD4, the host cell receptor for HIV [111,112]. Coupling of these nNAbs with CD4 enables competition for viral engagement, such that HIV binds to nNAb-conjugated CD4 rather than cell surface CD4. The combination of nNAb plus a CD4 binding-site mimetic is also reported to stabilise the availability of the otherwise hidden nNAb epitopes, highlighting the requirement for both the nNAb and CD4 for activity [112,113]. Intriguingly, these hybrid proteins were also demonstrated to recognise primary HIV-infected primary CD4^+^ T cells and stimulate ADCC against this population, with these activities enhanced relative to antibody alone or antibody plus soluble CD4, again highlighting the impact of linking antibody recognition to CD4 binding [112].

The concept of bifunctional antibodies has existed for many years [114] but has greatly increased in popularity with expanding clinical applications, particularly in cancer immunotherapy [115]. Lim et al. combined a nNAb targeting a more conserved region of the SARS-CoV-2 RBD with a synthetic, human VH-only NAb targeting spike to form a bispecific VH/Fab antibody [116]. This resulted in >20-fold increases in the potency of in vitro neutralisation of both pseudovirus and authentic SARS-CoV-2 relative to the neutralising IgG alone or cocktails of the reagents screened [116]. This increased potency is similar to the concept of enabling pairs discussed in Ebola infection, whereby increased activity of the NAb is observed in the presence of a nNAb [75].

Fc modification can also enhance antibody effectiveness. These modifications include glycosylation [117], site-directed mutagenesis (SDM), or multimerisation to enhance avidity [118,119]. An example of such modification is commonly applied to NAb to extend serum half-life by introducing point mutations that alter binding to Fc receptors (reviewed in [120]). Other Fc modifications, such as defucosylation/afucosylation, enhance ADCC activity through enhanced FcR binding [120]. Likewise, Moore et al. demonstrated that a combination of IgG1 Fc mutations, S267E, H268F, and S324T, in anti-CD20 mAb increased affinity for C1q, resulting in an up to 6.9-fold increase in vitro complement fixation and induced cytotoxicity against Burkitt’s lymphoma Ramos cells [121]. Such Fc modifications are quite well established, and whilst typically applied to NAb and cancer therapeutics, they could similarly be applied to increase the effectiveness and serum half-life of nNAbs. More innovative approaches include that of Barrock et al., who describe a tandem Fc format whereby IgG and IgA Fc regions are combined in tandem in a single antibody, enabling exploitation of the Fc effector functions of each subtype [122]. This includes IgG1, known to be a potent activator of NK-mediated cytotoxicity, inducing ADCC with a long half-life, and IgA2, with a known role in phagocytic cell engagement and induction of ADCP [122]. Whilst this study focused on cancer mAb therapy and not nNAbs, this technology might be exploited to enhance the antiviral effector functions of specific nNAbs. Combining 2 Fc regions may increase the breadth of Fc effector engagement relative to single Fc region monoclonals. Together, these methods of functionalisation may yield novel therapeutic strategies against viral infection.

## 7. Future Outlooks and Conclusions

Whilst neutralisation remains the gold standard for antiviral antibody activity, nNAbs should not be overlooked but instead viewed as potential adjunct agents of protection. nNAb activities are multifaceted and complex. This encompasses both advantageous and deleterious effects: mediating protection through ADCC, ADCP, AMCD, or epitope unmasking and exacerbation of infection through ADE and epitope masking. The protective capacity of nNAbs can vary depending on the context, antibody titres, or the presence of additional synergising antibodies. In contrast to tests of neutralisation, defining the mechanisms of nNAb activities may require more complex models and methods. As such, improvements in our capacity to assay protective nNAb functions are required. When developing new vaccines and antibody-based therapeutics, screening of non-neutralising functions should be included in efficacy assessments and candidate selection regimes. The potential for functionalisation through established or novel approaches will also allow the integration of favourable characteristics, representing a potential method to further diversify our arsenal in combatting viral infections.

## Figures and Tables

**Figure 1 antibodies-13-00028-f001:**
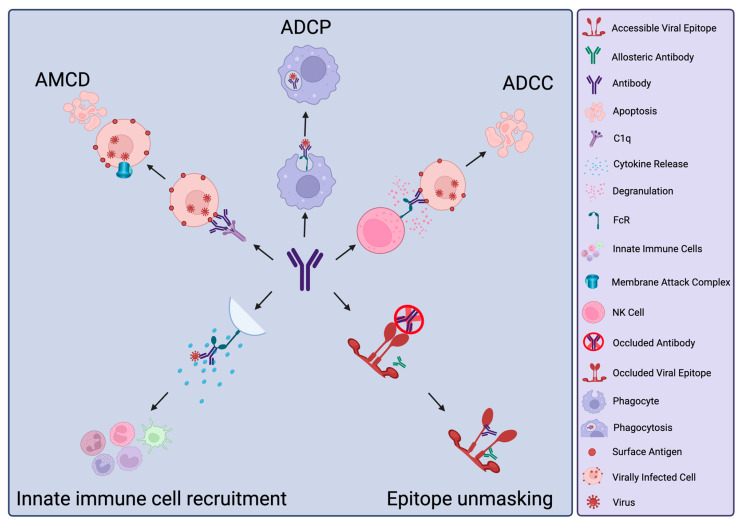
Non-neutralising antibody activities. Antibody-Mediated Complement Deposition (AMCD), Antibody-Dependent Cellular Phagocytosis (ADCP), Antibody-Dependent Cellular Cytotoxicity (ADCC). Key of icons is shown on the right. Created with Biorender.com, accessed 26 March 2024.

**Figure 2 antibodies-13-00028-f002:**
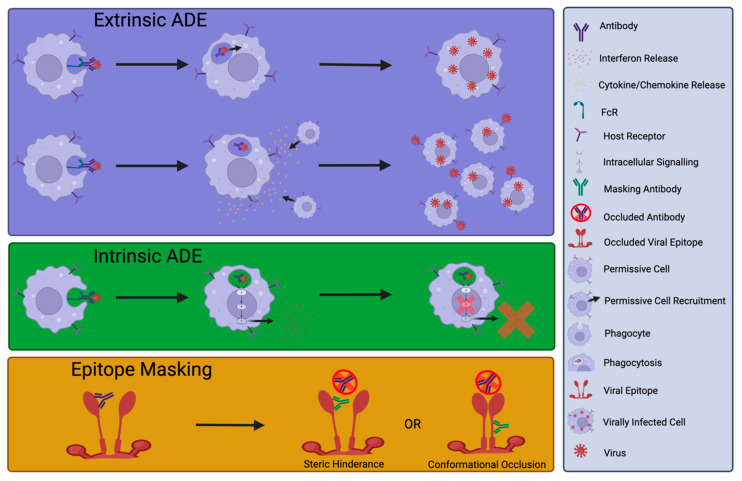
Simplified schematic of potential deleterious effects of antibody responses. Antibody Dependent Enhancement (ADE). Key of icons is shown on the right. Created with Biorender.com, accessed 26 March 2024.

**Figure 3 antibodies-13-00028-f003:**
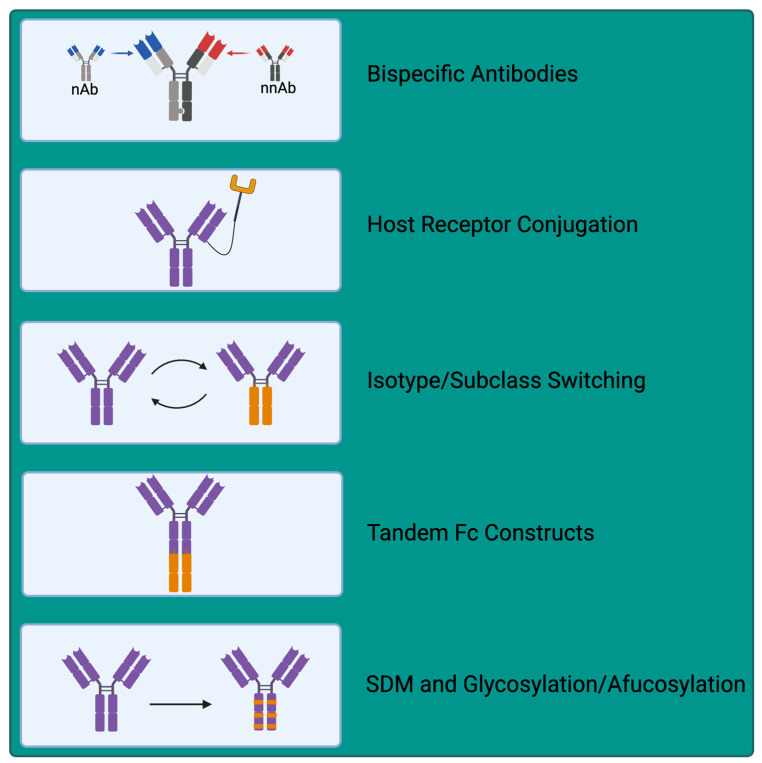
Simplified schematic depicting innovative potential functionalisation opportunities of non-neutralising antibodies. Site Directed Mutagenesis (SDM). Created with Biorender.com, accessed 26 March 2024.

## Data Availability

Not applicable.

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
