# Peer review of "Beyond bNAbs: Uses, Risks, and Opportunities for Therapeutic Application of Non-Neutralising Antibodies in Viral Infection"

_2073-4468, 2024, doi:10.3390/antib13020028_

Round 1

Reviewer 1 Report

Comments and Suggestions for Authors

The review by Mader K et al. provides a comprehensive summary of the effects and mechanisms of non-neutralizing antibodies in infectious settings. Their contributions are in line with current trends in the field and offer valuable insights into the subject matter. Additionally, the authors discuss the impacts and explore novel technologies for therapeutic purposes. The review is well-written, providing updated and informative content. Some minor corrections are suggested, including:

        Correcting "FcgRIIIA" instead of "IIA" in line 40.

•        Ensuring that Figures 1 and 3 are cited in the text.

•        Considering replacing "neutrophils" with "NK cells" as the main cell type performing ADCC (Figure 1).

•        Using "bNAbs" instead of "bnAb".

•        Clarifying the findings from Reference 42 regarding the production of chemokines and cytokines such as TNF and IL-6, as well as the detection of IFNb at the mRNA level (Figure 4 Panel F of ref 42) for accuracy in line 143.

•        Correcting "FcγR" instead of "FcyR" in line 237 and "FcγR" instead of "FcRγ" in line 256.

•        Adjusting "FcγRIIA" to "FcγRIIA" (and similarly for "FcγRIIB") in lines 313 and 314.

Additionally, the manuscript should include sections for Author Contributions, Funding, and Acknowledgments to comply with standard formatting requirements.

Author Response

We thank the reviewer for their careful reading of the manuscript and their encouraging comments. In response to the reviewer's suggestions, we have made the changes indicated as summarised below. All changes in the manuscript are highlighted in yellow.

  • Correcting "FcgRIIIA" instead of "IIA" in line 40.
    • We have corrected this on line 41 of the revised manuscript.
  • Ensuring that Figures 1 and 3 are cited in the text.
    • Thank you for catching this, the figures are now cited appropriately.
  • Considering replacing "neutrophils" with "NK cells" as the main cell type performing ADCC (Figure 1).
    • Thank you, this has been corrected in the revised figure 1
  • Using "bNAbs" instead of "bnAb".
    • corrected throughout the manuscript; we have also capitalised the N standing for Neutralisisng in NAb and nNAb for consistency.
  • Clarifying the findings from Reference 42 regarding the production of chemokines and cytokines such as TNF and IL-6, as well as the detection of IFNb at the mRNA level (Figure 4 Panel F of ref 42) for accuracy in line 143.
    • Corrected; please see lines 142-146 for revised text.
  • Correcting "FcγR" instead of "FcyR" in line 237 and "FcγR" instead of "FcRγ" in line 256.
    • Corrected.
  • Adjusting "FcγRIIA" to "FcγRIIA" (and similarly for "FcγRIIB") in lines 313 and 314.
    • We have kept the bold letters A and B in these words for clarity when comparing the effects of two receptors with very similar names (FcγRIIA vs. FcγRIIB).
  • Additionally, the manuscript should include sections for Author Contributions, Funding, and Acknowledgments to comply with standard formatting requirements.
    • The missing information has been added.

Reviewer 2 Report

Comments and Suggestions for Authors

I have read the review by Mader and Dustin. It is well-written, timely, and relevant. It is well structured and only a few things that need clarification or adjustment.

1.     Line 9. Perhaps keep references out of the abstract?

2.     Figure 1 would benefit from larger cells/drawings. Perhaps reduce the size of the arrows and enlarge the objects.

3.     Line 26-30. While true that neutralization is mostly through Fabs, for Sars-CoV-2 the Fc can sterically be important for neutralization by blocking the interaction between ace2 and RBD. This should be nuanced.

a.     Ref. Suryadevara N, Shrihari S, Gilchuk P, VanBlargan LA, Binshtein E, Zost SJ, Nargi RS, Sutton RE, Winkler ES, Chen EC, Fouch ME, Davidson E, Doranz BJ, Chen RE, Shi PY, Carnahan RH, Thackray LB, Diamond MS, Crowe JE Jr. Neutralizing and protective human monoclonal antibodies recognizing the N-terminal domain of the SARS-CoV-2 spike protein. Cell. 2021 Apr 29;184(9):2316-2331.e15. doi: 10.1016/j.cell.2021.03.029. Epub 2021 Mar 16. PMID: 33773105; PMCID: PMC7962591.

b.     McCallum M, De Marco A, Lempp FA, Tortorici MA, Pinto D, Walls AC, Beltramello M, Chen A, Liu Z, Zatta F, Zepeda S, di Iulio J, Bowen JE, Montiel-Ruiz M, Zhou J, Rosen LE, Bianchi S, Guarino B, Fregni CS, Abdelnabi R, Foo SC, Rothlauf PW, Bloyet LM, Benigni F, Cameroni E, Neyts J, Riva A, Snell G, Telenti A, Whelan SPJ, Virgin HW, Corti D, Pizzuto MS, Veesler D. N-terminal domain antigenic mapping reveals a site of vulnerability for SARS-CoV-2. Cell. 2021 Apr 29;184(9):2332-2347.e16. doi: 10.1016/j.cell.2021.03.028. Epub 2021 Mar 16. PMID: 33761326; PMCID: PMC7962585.

4.     Line 40 is an error, NK cells primarily perform ADCC through CD16 aka FcGR3A, not CD32A aka FcGR2A. Please find this ref as an example of this being the case:

a.     Wang W, Erbe AK, Hank JA, Morris ZS, Sondel PM. NK Cell-Mediated Antibody-Dependent Cellular Cytotoxicity in Cancer Immunotherapy. Front Immunol. 2015 Jul 27;6:368. doi: 10.3389/fimmu.2015.00368. PMID: 26284063; PMCID: PMC4515552.

5.     Line 64. That IgM does not have an FcG domain is obvious by definition, but since there is some crosstalk between different Fc receptors, where, for instance, FcA can interact with IgM to some extent, it is better to write that FcG receptors have low affinity for IgM.

6.     Line 78. Suggested rephrase “Antibodies do not all target” to “Antibodies do not necessarily target”.

7.     More general comment: The authors discuss the limitations of neutralization assays as correlates of protection, and that binding to antigen also is not sufficient to evaluate the immune response and vaccine response since it masks Fc-mediated effector functions. This is an important message in the review, and it would have benefited from referencing a few methods developed to screen Fc-mediated effector functions in a robust manner. A section of the review could be dedicated to this for ADCP, ADCC and ADCD where a few reference studies can be mentioned and discussed. A few suggestions on ADCP refs:

a.     Ackerman ME, Moldt B, Wyatt RT, Dugast AS, McAndrew E, Tsoukas S, Jost S, Berger CT, Sciaranghella G, Liu Q, Irvine DJ, Burton DR, Alter G. A robust, high-throughput assay to determine the phagocytic activity of clinical antibody samples. J Immunol Methods. 2011 Mar 7;366(1-2):8-19. doi: 10.1016/j.jim.2010.12.016. Epub 2010 Dec 27. Erratum in: J Immunol Methods. 2012 Feb 28;376(1-2):156. PMID: 21192942; PMCID: PMC3050993.

b.     de Neergaard T, Sundwall M, Wrighton S, Nordenfelt P. High-Sensitivity Assessment of Phagocytosis by Persistent Association-Based Normalization. J Immunol. 2021 Jan 1;206(1):214-224. doi: 10.4049/jimmunol.2000032. Epub 2020 Dec 2. PMID: 33268484.

Author Response

We thank the reviewer for their careful reading of the manuscript and their encouraging comments. We have made the changes requested (highlighted in yellow in the revised manuscript) and summarise our response to the reviewers comments here. Note that Figures 1 and 3 have been updated.

  1. Line 9. Perhaps keep references out of the abstract?
    • We have removed the citation from the abstract
  2. Figure 1 would benefit from larger cells/drawings. Perhaps reduce the size of the arrows and enlarge the objects.
    • Thank you for this suggestion; we provide a revised Figure 1 with enlarged cell icons and have also corrected our mistaken attribution of ADCC to neutrophils (as noted by Reviewer 1).
  3. Line 26-30. While true that neutralization is mostly through Fabs, for Sars-CoV-2 the Fc can sterically be important for neutralization by blocking the interaction between ace2 and RBD. This should be nuanced.
    • We have added the text "Such neutralisation may also include Fc-mediated steric hindrance [4,5]." at line 30, with citation of the recommended references (Suryadevara N et al, McCallum M et al) at numbers 4 and 5 in the bibliography. 

  4. Line 40 is an error, NK cells primarily perform ADCC through CD16 aka FcGR3A, not CD32A aka FcGR2A. Please find this ref as an example of this being the case:: Wang W et al....
    • Thank you for catching this error. We have made this correction in line 41 of the revised manuscript, and cite Wang et al in the bibliography (reference 9).

5. Line 64. That IgM does not have an FcG domain is obvious by definition, but since there is some crosstalk between different Fc receptors, where, for instance, FcA can interact with IgM to some extent, it is better to write that FcG receptors have low affinity for IgM.

    • This has been corrected (lines 65-66).

6. Line 78. Suggested rephrase “Antibodies do not all target” to “Antibodies do not necessarily target”.

    • Corrected (line 79).

7. More general comment: The authors discuss the limitations of neutralization assays as correlates of protection, and that binding to antigen also is not sufficient to evaluate the immune response and vaccine response since it masks Fc-mediated effector functions. This is an important message in the review, and it would have benefited from referencing a few methods developed to screen Fc-mediated effector functions in a robust manner. A section of the review could be dedicated to this for ADCP, ADCC and ADCD where a few reference studies can be mentioned and discussed. A few suggestions on ADCP refs:

  • Ackerman ME et al...
  • de Neergaard T et al...
    • Thank you for this excellent suggestion. We have added a section titled 'Measuring Fc-dependent effector functions' in lines 335-376. The suggested references (now references 105 and 106) are among those included in this section.